# *SDHA* Germline Mutations in SDH-Deficient GISTs: A Current Update

**DOI:** 10.3390/genes14030646

**Published:** 2023-03-04

**Authors:** Angela Schipani, Margherita Nannini, Annalisa Astolfi, Maria A. Pantaleo

**Affiliations:** 1Department of Medical and Surgical Sciences (DIMEC), University of Bologna, 40138 Bologna, Italy; 2Department of Pharmaceutics, Utrecht Institute for Pharmaceutical Sciences, Utrecht University, 3584 CS Utrecht, The Netherlands; 3Division of Oncology, IRCCS Azienda Ospedaliero-Universitaria di Bologna, 40138 Bologna, Italy

**Keywords:** succinate dehydrogenase complex (SDH), gastrointestinal stromal tumors, SDH-deficient GIST, *SDHA* germinal mutations

## Abstract

Loss of function of the succinate dehydrogenase complex characterizes 20–40% of all *KIT/PDGFRA*-negative GIST. Approximately half of SDH-deficient GIST patients lack *SDHx* mutations and are caused by a hypermethylation of the *SDHC* promoter, which causes the repression of *SDHC* transcription and depletion of SDHC protein levels through a mechanism described as epimutation. The remaining 50% of SDH-deficient GISTs have mutations in one of the SDH subunits and *SDHA* mutations are the most common (30%), with consequent loss of SDHA and SDHB protein expression immunohistochemically. *SDHB*, *SDHC*, and *SDHD* mutations in GIST occur in only 20–30% of cases and most of these SDH mutations are germline. More recently, germline mutations in *SDHA* have also been described in several patients with loss of function of the SDH complex. *SDHA*-mutant patients usually carry two mutational events at the *SDHA* locus, either the loss of the wild type allele or a second somatic event in compound heterozygosis. This review provides an overview of all data in the literature regarding *SDHA*-mutated GIST, especially focusing on the prevalence of germline mutations in SDH-deficient GIST populations who harbor *SDHA* somatic mutations, and offers a view towards understanding the importance of genetic counselling for *SDHA*-variant carriers and relatives.

## 1. Introduction

Gastrointestinal stromal tumors (GISTs) are mesenchymal tumors of the gastrointestinal tract deriving from the interstitial cells of Cajal (ICC), the mesenchymal cells responsible for the autonomous peristaltic contractions of the gut [1]. 

Around 10–15% of adult GISTs and all pediatric GISTs do not carry the driver mutations in the tyrosine kinase receptor KIT and/or platelet-derived growth factor receptor-α (PDGFR-α), and they have been called wild type GISTs (WT) [2]. Among the *KIT/PDGFRA* WT GISTs, about 15% carry mutations in the *RAS*-pathway (*BRAF/RAS* or *NF1*). On the other hand, between 20% and 40% are driven by the loss of function of succinate dehydrogenase (SDH), also defined as SDH-deficient GISTs. The remaining 50% are called quadruple WT (qWT) GISTs since they do not harbor any mutations in *KIT/PDGFRA/SDH/RAS-P* genes [3,4,5]. 

SDH-deficient GISTs tend to develop in children and young patients and can be sporadic or associated with Carney-Stratakis syndrome (CSS) and Carney triad (CT). Carney-Stratakis syndrome is a rare hereditary condition characterized by multifocal paragangliomas and GISTs. On the other hand, Carney triad is a rare non-hereditary disease causes by the association of GIST, pulmonary chondroma, and paraganglioma [6,7].

SDH-deficient GISTs exhibit characteristic clinical and pathological features, such as their primary localization in the stomach, multi-nodular growth, an epithelioid phenotype, and a common lymphovascular invasion [8]. This subset of GIST is known for the occurrence of lymph node metastases and an indolent course of the disease [9]. 

The immunohistochemical loss of succinate dehydrogenase subunit B (SDHB), which indicates the loss of function of the entire SDH-complex, is a representative feature of SDH-deficient GIST [10]. SDHB protein expression by immunohistochemistry is a diagnostic marker for loss of function of the SDH complex, whereas SDHA immunohistochemistry is able to identify *SDHA*-mutant GISTs [11]. SDH-deficient GISTs are uniformly immunohistochemically positive for KIT and DOG1/Ano1 (Anoctamin-1) [1]. They respond poorly to treatment with imatinib, suggesting that alternative signalling pathways are driving tumor formation [12]. SDH-deficient GISTs present an expression profile extremely different from *KIT*-mutant GISTs, showing the upregulation of distinctive gene pathways linked to tumor progression. Moreover, they display a depletion of immune competence, suggesting that this GIST subgroup can be considered a non-inflamed tumor [13].

SDH deficiency is mainly caused by the biallelic inactivation of one of the four SDH subunit genes (*SDHA*, *SDHB*, *SDHC*, and *SDHD*) according to the classic tumor suppressor gene model [14]. Normally, the combination of a first hit, that is, an inactivating germline mutation, with a second hit, represented by another inactivating mutation affecting the second allele or somatic loss of heterozygosity, leads to the loss of function of an SDHx subunit. Less frequently, SDH deficiency is caused by the somatic inactivation of both alleles of a given complex subunit or SDH assembly factor [14]. Approximately half of patients with *SDH*-mutated GISTs have mutations in one of the SDH subunits and *SDHA* mutations are the most common (30%) with consequent loss of SDHA and SDHB protein expression immunohistochemically [8,11]. However, it is widely recognized that the promoter hypermethylation and epigenetic silencing of the *SDHC* gene can result in the inactivation of the SDH complex [15,16]; germline mutations in *SDHB*, *SDHC*, and *SDHD* occur in only 20–30% of SDH-deficient GIST.

More recently, germline mutations in *SDHA* have also been reported in several patients with loss of function of the SDH complex. This review reports a current comprehensive state-of-the-art study on the germline *SDHA* mutations in SDH-deficient GIST populations who harbor *SDHA* somatic mutations by collecting and describing all studies in which these data have been reported.

## 2. SDH Complex in Cancers

Succinate dehydrogenase, SDH, is a mitochondrial enzyme complex (also known as complex II of the electron transport chain) that is involved in both the citric acid cycle, by catalysing the oxidation of succinate to fumarate, and the electron transport chain, by leading the electron transfer to the terminal acceptor ubiquinone (Figure 1a) [17]. The assembled SDH complex, located in the inner membrane of the mitochondria, consists of four subunits encoded by nuclear genes (*SDHA*, *SDHB*, *SDHC* and *SDHD*, collectively indicated as *SDH*x). All of the *SDH* genes are involved in the tumorigenesis of different types of cancers including GIST, paraganglioma, pheochromocytoma, chronic lymphocytic leukemia, renal cell carcinoma, thyroid cancer, Hodgkin lymphoma, pituitary adenomas, and neuroendocrine tumors of the pancreas [18,19,20,21,22].

## 3. *SDHA* Alterations in GIST

Tumor development caused by SDH deficiency results from the complete loss of function of one SDHx subunit, which leads to instability of the entire SDH complex and loss of the enzymatic function of the SDH complex. Disruption of the succinate dehydrogenase complex (SDH), caused by either biallelic mutations or epigenetic silencing, leads to accumulation of succinate, an oncometabolite that promotes tumorigenesis by activating different pathways. Accumulation of succinate leads to the inhibition of prolyl-hydroxylase domain proteins (PHD) activity, causing the stabilization of hypoxia-inducible factor 1 (HIF1), which in turn leads to increased expression of the transcriptional factors as vascular endothelial growth factor (VEGF) and insulin-like growth factor-1 (IGF1), causing increased angiogenic and cell growth signalling (Figure 1b). SDH-deficient GISTs do not show a *IGF1R* genomic amplification; however, the IGF-1 receptor (IGF1R) is overexpressed both at the mRNA and protein level [23]. 

Moreover, the accumulation of succinate, which has a similar structure of α-ketoglutarate (α-KG), leads to the inhibition of the α-KG dependent dioxygenases such as JmjC domain containing histone lysine demethylases (KDM) and ten–eleven translocation (TET) enzymes (Figure 1b). The latter are involved in DNA demethylation and transcriptional silencing because they drive the active elimination of 5-methylcytosine (5mC) from methylated CpG sites. Consequently, genome-wide DNA hypermethylation may be related to the disruption of DNA demethylation machinery by downregulated TET enzyme [24]. Intriguingly, it has been recently shown that the characteristic hypermethylation in these tumors is associated with changes in genome topology that can drive an oncogenic program. The hypermethylation is associated with pervasive insulator losses and topological reorganization of the *FGF* and *KIT* loci. In particular, the hypermethylated phenotype interrupts the binding of CTCF in regions located in proximity to the *FGF3*/*FGF4* locus, leading to *FGF4* overexpression [25]. 

## 4. Germline *SDH* Mutations in GIST

In most SDH-deficient GISTs, the loss of the complex is caused by the presence of mutations in at least one of the four subunits. Mutations in all four *SDH* genes have been described, but different studies have classified *SDHA* as the most frequently mutated subunit, with *SDHA* mutations occurring in ~30% of all SDH-deficient GIST [26]. SDHA patients usually carry two mutational events at the *SDHA* locus, either the loss of the wild type allele or a second somatic event in compound heterozygosis. *SDHB*, *SDHC* and *SDHD* mutations in GIST are less frequent (20–30%) and most of these mutations are germline [19,27]. Alongside germline mutations of *SDHB*, *SDHC* and *SDHD*, evidence of germline *SDHA* has been accumulated.

### SDHA Germline Mutations

Notably, numerous findings have evaluated that germline mutations in *SDHA* are highly frequent in SDHA-deficient GIST (Table 1). In 2013, Dwight et al. [28] investigated a sample of 10 SDH-deficient GISTs and displayed that 30% of SDH-deficient GISTs in their series were associated with germline SDHA mutations.

Additionally, Oudijk et al. performed SDHA IHC on 33 WT-GISTs, including nine pediatric/adolescent GISTs, in order to study whether IHC could identify *SDHA*-mutated GISTs. SDHA-negative samples were sequenced both in corresponding tumor and germline DNA isolated from paraffin-embedded healthy tissue surrounding the tumor; four cases were germline [29].

In a cohort of 11 SDHB-negative WT GIST, five patients (45%) carried *SDHA* mutation; among them, three patients for which germline DNA was available harbored germline mutation. The presence of a germline mutation accompanied by loss of the allele in the tumor is presumed also in the two patients for which germline DNA was not available; based on tumor sequencing, the data revealed the presence of homozygous missense *SDHA* mutations [30].

Moreover, heterozygous mutations were observed in DNA from normal tissue of all six out of nine SDHA-deficient GIST patients whose material was available [31]. 

Furthermore, in a 2016 study, a large sample set of 95 patients was investigated, in which 84 had succinate dehydrogenase-deficient GIST caused either by *SDH* mutations (75%) or by *SDHC* promoter hypermethylation (25%), and 18 were syndromic GISTs with chondromas and/or paragangliomas. Of the 63 cases of *SDH*-mutant GISTs, 16 had mutation in *SDHB*, 12 in *SDHC*, and 1 in *SDHD*, and 34 had mutations in *SDHA*; among them, 20 cases were germline [32]. 

*SDHA* mutations have been observed in an additional seven SDHA/B-deficient cases and six of them were germline mutations [11]. In addition, Carrera et al. identified LOH at the *SDHA* locus in one GIST patient [33].

The frequency of germline *SDHA* mutations within the SDH-deficient GISTs is even higher in a recent study that identifies germline mutations in all 14 patients analyzed for which a normal counterpart was available [34,35,36].

**Table 1 genes-14-00646-t001:** List of *SDHA* germline mutations reported in SDH-deficient GISTs.

ID	Age	Sex	Cytology	Site	Associated Tumor	ICH SDHB	IHC SDHA	Exon	Germline Mutation (cDNA, Protein)	References
#1	22	M	-	stomach	no	neg	neg	2	c.91C>T; p.R31X	Italiano et al., 2012 [37]
#2	31	M	mixed	stomach	-	neg	neg	5	c.553C>T; p.Q185X	Wagner et al., 2013 [31]
#3	38	F	epithelioid	stomach	-	neg	neg	11	c.1534C>T; p.R512X	Wagner et al., 2013
#4	39	F	mixed	stomach	-	neg	neg	8	c.688delG; Frameshift	Wagner et al., 2013
#5	22	M	mixed	stomach	-	neg	neg	2	c.91C>T; p.R31X	Wagner et al., 2013
#6	53	F	mixed	stomach	-	neg	neg	2	c.91C>T; p.R31X	Wagner et al., 2013
#7	19	M	mixed	stomach	-	neg	neg	2	c.91C>T; p.R31X	Wagner et al., 2013
#8	45	F	epithelioid	stomach	no	neg	neg	15	c.1969G>A; p.V657I	Dwight et al., 2013 [28]
#9	25	F	epithelioid	stomach	no	neg	neg	3	c.160C>T; p.Q54X	Dwight et al., 2013
#10	41	M	epithelioid	stomach	no	neg	neg	12	c.1633+3G>C; splice-site	Dwight et al., 2013
#11	39	F	epitheliod/spindle	stomach	no	neg	-	7	c.818C>T; p.T2731I	Belinsky et al.,2013 [30]
#12	52	F	epitheliod/spindle	stomach	no	neg	-	IVS4/ex5	c.457-2_c457delAGC; p.L153Kfs*71	Belinsky et al.,2013
#13	33	F	epitheliod/spindle	stomach	no	neg	-	2	c.91C>T; p.R31X	Belinsky et al.,2013
#14	41	F	epithelioid	stomach	MTC	neg	neg	2	c.91C>T; p.R31X	Oudijk et al., 2013 [29]
#15	53	F	spindle cell	stomach	no	neg	neg	2	c.91C>T; p.R31X	Oudijk et al., 2013
#16	47	F	mixed	stomach	no	neg	neg	2	c.91C>T; p.R31X	Oudijk et al., 2013
#17	14	M	mixed	stomach	no	neg	neg	2	c.91C>T; p.R31X	Oudijk et al., 2013
#18	-	-		stomach	no	neg	neg	2	c.91C>T; p.R31X	Miettinen et al., 2013 [11]
#19	-	-		stomach	no	neg	neg	2	c.91C>T; p.R31X	Miettinen et al., 2013
#20	-	-		stomach	no	neg	neg	2	c.91C>T; p.R31X	Miettinen et al., 2013
#21	-	-		stomach	no	neg	neg	6	c.767C>T; p.T256I	Miettinen et al., 2013
#22	-	-		stomach	no	neg	neg	13	c.1794G>C; p.K598N	Miettinen et al., 2013
#23	-	-		stomach	no	neg	neg	14	c.1795–1G>T; Ex 14 5’ splicing	Miettinen et al., 2013
#24	-	-		stomach	no	neg	pos	5	c.562C>T; p.R188W	Miettinen et al., 2013
#25	23	M	epitheliod	stomach	RCC	neg	pos	1	c.2T>C; p.M1T	Jiang et al., 2015 [37]
#26	-	-	-	stomach	no	neg		10	c.1432_1432del; p.478_478del	Boikos et al., 2016 [32]
#27	-	-	-	stomach	no	neg	-	11	c.1513delA; p.S505fs	Boikos et al., 2016
#28	-	-	-	stomach	no	neg	-	14	c.1795-1G>T	Boikos et al., 2016
#29	-	-	-	stomach	no	neg	-	10	c.1340°>G; p.H447R	Boikos et al., 2016
#30	-	-	-	stomach	no	neg	-	10	c.1367C>T; p.S456L	Boikos et al., 2016
#31	-	-	-	stomach	no	neg	-	13	c.1753C>T; p.R585W	Boikos et al., 2016
#32	23	F	epitheliod	stomach	CHO	neg	-	3	c.295C>T; p.H99Y	Boikos et al., 2016
#33	-	-	-	stomach	no	neg	-	5	c.562C>T; p.R188W	Boikos et al., 2016
#34	-	-	-	stomach	no	neg	-	7	c.818C>T; p.T273I	Boikos et al., 2016
#35	-	-	-	stomach	no	neg	-	2	c.91C>T; p.R31X	Boikos et al., 2016
#36	-	-	-	stomach	no	neg	-	2	c.91C>T; p.R31X	Boikos et al., 2016
#37	21	M	epitheliod/spindle	stomach	CHO	neg	-	2	c.91C>T; p.R31X	Boikos et al., 2016
#38	-	-	-	stomach	no	neg	-	2	c.91C>T; p.R31X	Boikos et al., 2016
#39	-	-	-	stomach	no	neg	-	2	c.91C>T; p.R31X	Boikos et al., 2016
#40	-	-	-	stomach	no	neg	-	2	c.91C>T; p.R31X	Boikos et al., 2016
#41	-	-	-	stomach	no	neg	-	2	c.91C>T; p.R31X	Boikos et al., 2016
#42	-	-	-	stomach	no	neg	-	8	c.923C>T; p.T308M	Boikos et al., 2016
#43	-	-	-	stomach	no	neg	-	13	c.1794G>C; p.K598N	Boikos et al., 2016
#44	30	M	epitheliod/spindle	stomach	PGL, CHO	neg	-	11	c.1532T>C; p.L511P	Boikos et al., 2016
#45	-	-	-	stomach	-	neg	-		SDHA deletion	Boikos et al.,2016
#46	21	-	-		-	neg	neg	2	c.91C>T; p.R31X	Gault et al., 2018 [38]
#47	27	-	-		-	neg	neg	2	c.91C>T; p.R31X	Gault et al., 2018
#48	20	F	epithelioid	stomach	PGL	-	-	1	c.1A > C; p.(Met1?)	Carrera et al., 2019 [33]
#49	28	F		stomach	no			9	c.1151 C>G; p.S384X	Pantaleo et al., 2011 [26]
#50	30	M		stomach	no			2	c.91 C>T; p.R31X	Pantaleo et al., 2011
#51	39	F		stomach	no			13	c.1766 G>A; p.R589Q	Pantaleo et al., 2011 [34]
#52	37	F		stomach	no			5	c.457-3_457-1 delCAG; p.L153splice	Pantaleo et al., 2014 [35]
#53	31	F		stomach	no			9	c.1151 C>G; p.S384X	Pantaleo et al., 2022 [36]
#54	61	M		stomach	PGL			6	c.698G>T; p.G233V	Pantaleo et al., 2022
#55	21	F		stomach	no			5	c.512G>A; p.R171H	Pantaleo et al., 2022
#56	38	M		stomach	no			6	c.768G>C; p.G257A	Pantaleo et al., 2022
#57	70	F		stomach	no			4	c.356G>A; p.W119X	Pantaleo et al., 2022
#58	66	F		stomach	no			14	c.1799G>A; p.R600Q	Pantaleo et al., 2022
#59	17	M		stomach	no			12	c.1663+3 G>C; p.R554splice	Pantaleo et al., 2022
#60	55	F		stomach	no			14	c.1799G>A; p.R600Q	Pantaleo et al., 2022
#61	50	M		stomach	no			13	c.1754G>A; p.R585Q	Pantaleo et al., 2022
#62	17	M		stomach	no			6	c.628C>T; p.R210X	Pantaleo et al., 2022

## 5. *SDHA* Germline Consideration 

Taken together, these data demonstrate that the most common mutation in the analyzed patient cohort occurs in exon 2 of *SDHA* gene, c.91C>T (Figure 2). This sequence change produces a premature translational stop signal at codon 31 (p.R31X) of the *SDHA* gene, leading to a truncated protein [8,11,30,37,38].

Interestingly, the high frequency of this mutation in different series suggests the presence of a founder effect, but the incidence of the *SDHA* p.R31X mutation in different countries makes such founder mutation less likely and might suggest a hotspot mutation. 

Another important consideration is the recurrence of the germline *SDHA* mutation in other cancers. Based on the previously mentioned studies, we can affirm that most *SDHA* mutations in GISTs are GIST-specific. An association exists between germline *SDHA* mutations and GIST onset, supporting the idea that GISTs develop alone as the only tumor disease due to *SDHA*, and not in association with other tumors. To date, only a few cases report *SDHA* germline mutation in the context of other tumors: paraganglioma in one case, pulmonary chondroma in two cases, and the full Carney triad in one patient [33,38,39].

Additionally, the loss of SDHA expression in tumors consistently predicts the presence of *SDHA* mutations in tumor cells and associated germline material. Owing to the fact that most of the cases are SDHA-negative at the protein level, the possibility to use immunohistochemistry as a reliable tool for identifying *SDHA*-mutant GISTs may be considered. All the authors have also clearly showed that the predominant tumor site is the stomach in the totality of cases.

Concerning the issue of the pathogenic effect of different mutations, it is known that genetic variants are usually classified following the American College of Medical Genetics and Genomics (ACMG) guidelines in a five-tiered scheme of pathogenic, likely pathogenic, variant of uncertain significance (VUS), likely benign, or benign. With the increased use of molecular testing and next-generation sequencing (NGS), many *SDHA* variants of unknown significance (VUS) have been detected. The meaning of VUS is controversial since it is unclear if the variant is related to the disease or not. Up to now, the usage of VUS data in the clinical context is challenging in term of patient management and treatment decisions, and there is a wide agreement on the fact that a VUS should not be used in clinical decision-making. Likewise, if a patient is identified to have a VUS, all clinical decisions should not be based on the presence of the VUS but only on personal and family history.

## 6. Other SDH Subunit Mutations

SDH-deficient GISTs are the majority of WT GISTs, with germline *SDHA* mutations as the most frequent mutational event [32,34]. The relative frequency of germline mutations in the SDH subunits has been clearly explored in an observational study of WT GISTs conducted in a large series of 84 SDH-deficient cases. Within the 63 cases of SDH-mutant GIST, 54% had mutations in *SDHA* subunit, mostly germline, while *SDHB/C/D* altogether represented the remaining 46% of SDH-mutant GIST [32]. Moreover, while only a minority of GISTs with *SDHA* mutations (nine of twenty-five available samples) had loss of heterozygosity as the second hit, most tumors with *SDHB* mutations (12 of 13 available samples) showed loss of heterozygosity as the second oncogenic event [32]. Wild type GIST driven by germline mutations in *SDHB*, *SDHC*, or *SDHD* subunits is a component of the Carney-Stratakis syndrome, an inherited predisposition syndrome characterized by the presence of GIST and paraganglioma [40]. In the same study, 22% of SDH-deficient tumors had hypermethylation of the *SDHC* promoter sequence [32], leading to loss of SDHC protein expression and functional SDH deficiency [27,41]. These SDH-epimutant tumor patients were younger, overwhelmingly female, had disease of gastric origin, and were often metastatic [32].

The same scenario is evident also in Miettienen et al. [11], who investigated 21 SDH-mutant GISTs, finding 10 *SDHA* mutations and 11 cases carrying either *SDHB*, *SDHC,* or *SDHD* mutations, mostly germline.

Overall, patients with *SDHA*-mutant tumors tend to have an older median age compared to the SDH-deficient GISTs due to other subunit mutations [36,42], while *SDHB*, *SDHC*, and *SDHD* mutant GISTs usually also develop paragangliomas [34,43].

Even if germline testing is recommended for all SDH-deficient GISTs, currently, there are no specific surveillance guidelines for these patients, and follow-up should parallel that of the general GIST population.

## 7. Current Therapies of SDH-Deficient GISTs

To date, the medical management of SDH-deficient GISTs is still controversial because of limited data available, both due to the rarity of this molecular subset of GIST and to the lack of SDH deficiency characterization in most studies [44]. Generally, given the lack of gain-of-function *KIT* and *PDGFRA* mutations, SDH-deficient GISTs are widely considered resistant to all tyrosine kinase inhibitors (TKIs), as all other *KIT/PDGFRA* WT GIST. Thus, there is a consensus to avoid imatinib or any adjuvant treatment in this rare molecular subset of GIST [45]. 

In the metastatic setting, there are few retrospective data showing some degree of activity of TKIs with anti-angiogenic properties, such as sunitinib, regorafenib, and pazopanib, likely because of the pseudohypoxic phenotype of SDH-deficient GIST [32,46,47,48]. 

Recent advances on the molecular background of SDH-deficient GISTs have shifted the therapy focus from the standard TKIs to other therapeutic strategies. In particular, given the known overexpression of IGF1R of SDH-deficient GISTs, the oral IGF-1R TKI linsitinib has been tested in a phase II study on adult and pediatric patients with wild type GIST, including 15 SDH-deficient GIST, showing a clinical benefit rate (CBR) and progression-free survival (PFS) at 9 months of 40% and 52%, respectively, suggesting a potential benefit of linsitinib in this patient population, though no objective responses were revealed [49]. 

Moreover, the O6-methylguanine-DNA methyltransferase (MGMT) promoter methylation markedly prevalent in SDH-deficient GISTs has represented some proof of concept about the potential role of alkylating agents in this subset of GISTs [50], whereas a phase II trial on temozolomide in advanced SDH-GISTs is still ongoing (NCT03556384); a prolonged disease stability after 18 consecutive cycles of temozolomide has been recently reported in a female metastatic and progressive SDH-deficient GIST 45. The same authors presented another anecdotal case with avid disease on DOTATATE PET, experiencing a durable partial response to 177lutetium-DOTA-octreotate PRRT after failing two lines of TKI therapy [51]. 

Finally, in a recent phase Ib trial, the combination of imatinib and binimetinib has shown encouraging activity in SDH-deficient GIST cohort, suggesting a potential role of targeting the ICC/GIST lineage-specific survival transcription factor, ETV1 [52].

In this evolving scenario, all data should be carefully interpreted in relation to the known frequent indolent behavior of SDH-deficient GISTs which could be underlying most prolonged disease stabilities and the clinical benefit observed in almost all patients.

## 8. Conclusions

In this review, we have collected data from the literature in order to present the current update of *SDHA* germline mutations occurring in GISTs. These studies together suggest that there is a high incidence of germline mutations in *SDHA***-**deficient GISTs harboring *SDHA* somatic mutations. Since the majority of *SDHA***-**mutated GISTs have germline mutations, the possibility of genetic counselling of *SDHA* mutation carriers and relatives should be evaluated.

## Figures and Tables

**Figure 1 genes-14-00646-f001:**
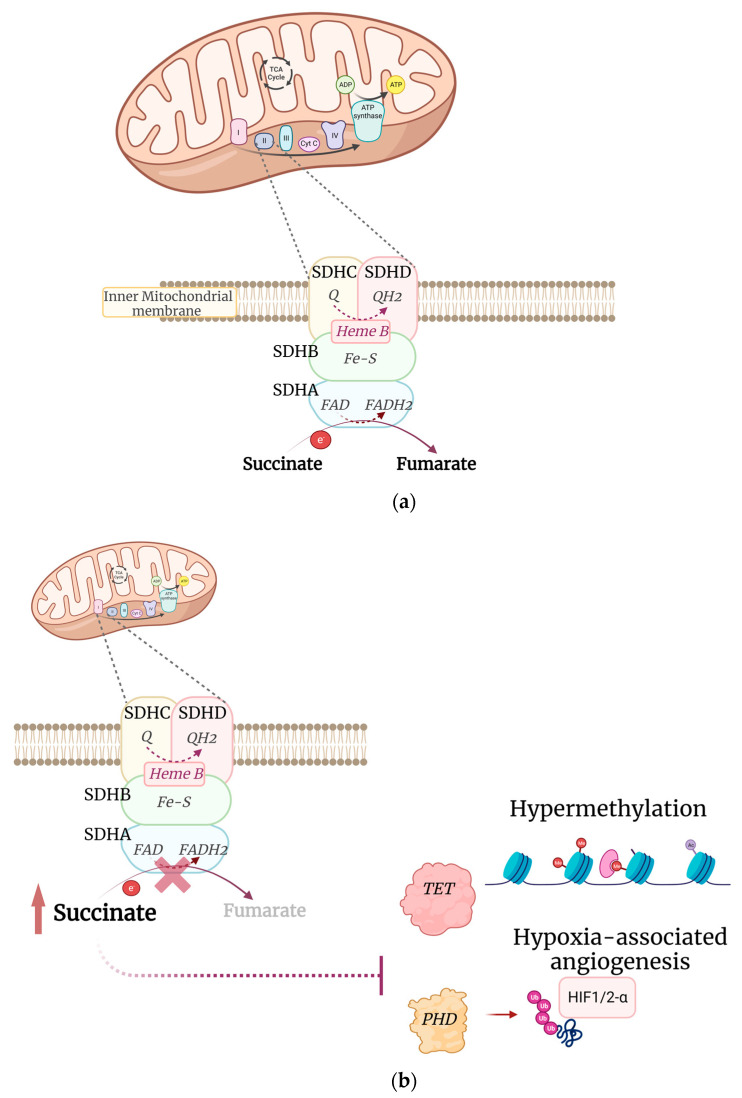
Schematic representation of the SDH complex in physiological (**a**) and pathological (**b**) condition. SDH, succinate dehydrogenase; Q,ubiquinone; QH2,Ubiquinol; Fe–S, iron-sulfur; FAD, flavin adenine dinucleotide; FADH2, dihydroflavine-adenine dinucleotide; e, electron; TET, ten–eleven translocation; Me, methyl group; PHD, prolyl-hydroxylase domain proteins; Ub, ubiquitin; HIF, hypoxia-inducible factor.

**Figure 2 genes-14-00646-f002:**
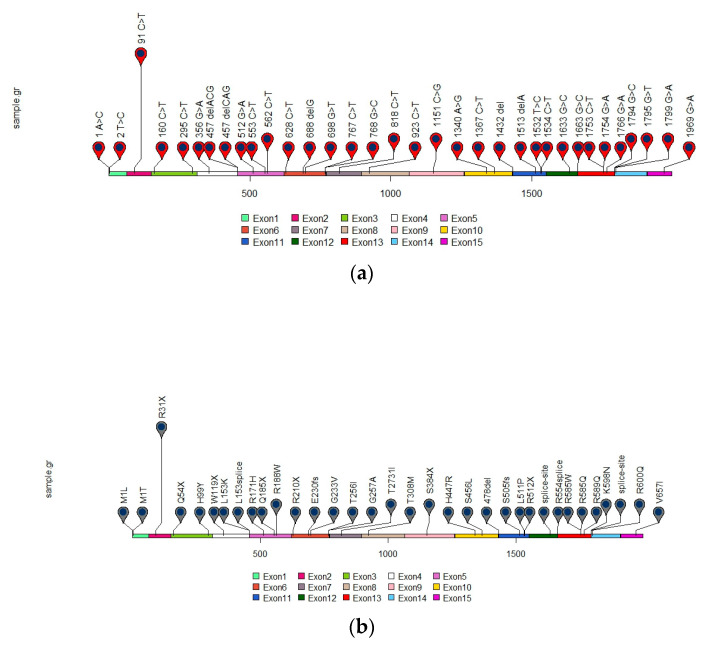
Distribution of reported *SDHA* gene mutations in GIST. Mutations are annotated at the cDNA (**a**) and protein level (**b**).

## Data Availability

Data sharing not applicable.

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
