# Peer review of "SDHA Germline Mutations in SDH-Deficient GISTs: A Current Update"

_genes, 2023, doi:10.3390/genes14030646_

Round 1

Reviewer 1 Report

The manuscript is well written and interesting.

I think is necessary to have a list of abbreviation (including the genes and their names).

The author must add a figure with SDH complex and his role for better clarity.

The author must add some information regarding the variants of unknown significance in SDHA gene and the genetic counseling and management in prevention of GIST in these cases. 

Author Response

Reviewer 1

The manuscript is well written and interesting.

> We thank the reviewer for this appreciation.

I think is necessary to have a list of abbreviation (including the genes and their names).

> As suggested by reviewer 1, a list of abbreviations was added on page 10, lines 304-314.

The author must add a figure with SDH complex and his role for better clarity.

> A novel figure (Figure 1) representing the biochemical structure and function of the SDH complex has been added on page 3 – 4, as requested.

The author must add some information regarding the variants of unknown significance in SDHA gene and the genetic counseling and management in prevention of GIST in these cases.

> Regarding the variants of unknown significance, we added a comment within paragraph 5 (page 6, lines 217-227).

Reviewer 2 Report

In this review, the authors simply list the relevant published literature on SDHA germline mutations occurring in GISTs, with less basic information and important comparative data, and then claim a high incidence of germline mutations in SDHA-deficient GISTs carrying SDHA somatic mutations. As the authors point out in the review, SDHA is the most common mutant subunit, but most SDHB, SDHC, and SDHD mutations are germline mutations. In addition, this review provides less information on the importance and benefits of genetic counseling for SDHA variant carriers and relatives. Even more, in this review, there are fewer data comparing SDHA variants with other variants such as SDHB, SDHC and SDHD, such as importance in diagnosis, treatment, outcome and genetic counseling. This review would have been more substantive and clinically relevant if the authors had provided more information on the above.

Author Response

Reviewer 2

In this review, the authors simply list the relevant published literature on SDHA germline mutations occurring in GISTs, with less basic information and important comparative data, and then claim a high incidence of germline mutations in SDHA-deficient GISTs carrying SDHA somatic mutations. As the authors point out in the review, SDHA is the most common mutant subunit, but most SDHB, SDHC, and SDHD mutations are germline mutations. In addition, this review provides less information on the importance and benefits of genetic counseling for SDHA variant carriers and relatives. Even more, in this review, there are fewer data comparing SDHA variants with other variants such as SDHB, SDHC and SDHD, such as importance in diagnosis, treatment, outcome and genetic counseling. This review would have been more substantive and clinically relevant if the authors had provided more information on the above.

> As correctly inferred by the reviewer, the matter of SDH mutations with respect to diagnosis, treatment, outcome and genetic counseling is very complex, and it is further complicated by the rarity of the condition and the disease. Anyway, to address the issue raised by the reviewer we added a paragraph entitled “Other SDH subunit mutations”, dealing with mutation frequency and clinical features of different SDH subunit mutations (page 6, line 229 – page 7, line 253).

Round 2

Reviewer 2 Report

The manuscript is a significant improvement over the previous version, with the addition of a schematic diagram of the mechanism of action of the SDH complex in physiological and pathological states and information on other SDH subunit mutations. The authors have responded appropriately to most of the questions posed by the reviewer. However, the writer still needs to pay attention to the consistency of the paragraph frame structure (some paragraphs are too far apart, while others are too close together). Thus, authors need to go through the manuscript carefully and pay more attention to typos, spaces, and grammar.